# Univariable and multivariable mendelian randomization study revealed the modifiable risk factors of urolithiasis

**Hailin Fang**[☯], **Jiwang Deng**[☯], **Qingjiang Chen, Dong Chen, Pengfei Diao, Lian Peng, Bin Lai, Yongmao Zeng, Yuefu Han**[ORCID]*

Department of Urology, Yuebei People's Hospital, Shantou University Medical College, Shaoguan, China

☯ These authors contributed equally to this work.
* hyf2540@126.com

**Data Availability Statement:** All GWAS files are available from the IEU open gwas project, https://gwas.mrcieu.ac.uk.

## Abstract

### Background

Urolithiasis is a common urological disease with increasing incidence worldwide, and preventing its risk poses significant challenges. Here, we used Mendelian randomization (MR) framework to genetically assess the causal nature of multifaceted risk factors on urolithiasis.

### Methods

17 potential risk factors associated with urolithiasis were collected from recently published observational studies, which can be categorized basically into lifestyle factors and circulating biomarkers. The instrumental variables of risk factors were selected from large-scale genome-wide association studies (N ≤ 607,291). Summary-level data on urolithiasis were obtained from UK Biobank (UKB) (3,625 cases and 459,308 noncases) and the FinnGen consortium (5,347 cases and 213,445 noncases). The univariable and multivariable MR analyses were applied to evaluate the causal, independent effect of these potential risk factors upon urolithiasis. Effects from the two consortia were combined by the meta-analysis methods.

### Results

Higher genetically predicted sex hormone-binding globulin (SHBG, OR, 0.708; 95% CI, 0.555 to 0.903), estradiol (OR, 0.179; 95% CI, 0.042 to 0.751), tea intake (OR, 0.550; 95% CI, 0.345 to 0.878), alcoholic drinks per week (OR, 0.992; 95% CI, 0.987 to 0.997), and some physical activity (e.g., swimming, cycling, keeping fit, and bowling, OR, 0.054; 95% CI, 0.008 to 0.363) were significantly associated with a lower risk of urolithiasis. In the Multivariate Mendelian Randomization (MVMR) analyses, the significant causal associations between estradiol, SHBG, tea intake, and alcoholic drinks per week with urolithiasis were robust even after adjusting for potential confounding variables. However, the previously observed causal association between other exercises and urolithiasis was no longer significant after adjusting for these factors.

**Funding:** The author(s) received no specific funding for this work.

**Competing interests:** The authors have declared that no competing interests exist.

## Conclusions

The univariable and multivariable MR findings highlight the independent and significant roles of estradiol, SHBG, tea intake, and alcoholic drinks per week in the development of urolithiasis, which might provide a deeper insight into urolithiasis risk factors and supply potential preventative strategies.

## Introduction

Urolithiasis is a prevalent urological disease with an increasing trend of global prevalence over the past few decades [1]. Due to the high prevalence of urolithiasis, patients with kidney stones without proper treatment suffer much pathogenic pain, including renal colic, urinary tract infections, painful urination, vomiting, and functional damage of the kidneys [2]. Moreover, urolithiasis imposes a tremendous strain on public healthcare systems, with the annually expenditure on kidney stone treatments in the USA alone exceeding 2 billion dollars [3]. The formation of kidney stones is a multifactorial and complex process linked to various risk factors, like genetic variation, circulating biomarkers, dietary habits, lifestyle, and living environment [4, 5]. However, despite these insights, current health interventions struggle to effectively mitigate the risk of urolithiasis, mainly due to the lack of clarity surrounding the causal relationship between these risk factors and the development of urolithiasis. This highlights the urgent need to identify the independent causal contributors to urolithiasis in order to alleviate its associated disease burden.

Circulating biomarkers and lifestyle factors have been demonstrated to be identified as key variables closely associated with the occurrence of kidney stones. An observational study found that sex hormones (including testosterone, dihydrotestosterone, estradiol, and SHBG were significantly different in kidney stone formers in comparison to the healthy population, which may help explain the higher incidence in males than in females [6]. However, the specific association between those sex hormones and kidney stones is unknown. When it comes to lifestyle factors, research indicates a possible connection between smoking and renal stones, although the association between alcohol and physical activity is less clear [7]. Additionally, the impact of tea and coffee on kidney stones remain controversial [8, 9]. Notably, these associations between modifiable exposures and kidney stones are demonstrated in observational research to sometimes be confounded and, thus, misleading, including reverse causation bias. Thus, confirming the genuine causal risk factors of kidney stones becomes a critical prerequisite for future disease prevention and treatment and strategies. The MR approach with genetic markers as an instrumental variable (IV) has been widely used to assess the causal relationships of the exposure on the outcome in epidemiology [10–12]. MR could not only minimize confounding by environmental factors as allelic genes are both randomly assigned and fixed during gamete formation and conception, but also overcome reverse causation bias as allelic randomization antedates the onset of disease.

In the current study, we first collected 17 potential risk factors associated with urinary stones from recently published observational studies. After that, we investigated the causal and independent link between urolithiasis and their risk factors from the UK Biobank (UKB) and FinnGen cohorts, especially circulating biomarkers and lifestyle factors, using MR framework and meta-analyses.

## Materials and methods

### Study design

The two-sample MR design, an IV analysis, was applied to strengthen the inference on the causal effect of exposures upon outcomes by exploiting genetic variants as IVs of exposure

| | UK Biobank | | | | FinnGen | | |
|---|---|---|---|---|---|---|---|
| | IVW | MRE | WM | | IVW | MRE | WM |
| **Circulating proteins** | | | | | | | |
| Estradiol levels | | | | | ↓ | ↓ | ↓ |
| SHBG | | | | | ↓ | ↓ | ↓ |
| Total testosterone levels | | | | | ↑ | ↑ | ↑ |
| PTH | ↓ | ↑ | ↑ | | ↓ | ↑ | ↓ |
| CRP | ↑ | ↑ | ↑ | | ↓ | ↓ | ↓ |
| IL6 | ↓ | ↑ | ↑ | | ↑ | ↑ | ↑ |
| IL18 | ↑ | ↑ | ↑ | | ↓ | ↓ | ↓ |
| IL27 | ↑ | ↑ | ↑ | | ↑ | ↑ | ↓ |
| IL8 | ↓ | ↓ | ↓ | | ↓ | ↓ | ↓ |
| IL16 | ↓ | ↓ | ↓ | | ↓ | ↓ | ↓ |
| IL1Ra | ↑ | ↑ | ↑ | | ↑ | ↑ | ↑ |
| **Lifestyle Factors** | | | | | | | |
| Tea intake | | | | | ↓ | ↓ | ↓ |
| Garlic intake | | | | | ↓ | ↑ | ↑ |
| Coffee consumption | ↓ | ↓ | ↓ | | ↓ | ↓ | ↓ |
| Age Of Smoking Initiation | ↓ | ↑ | ↓ | | ↑ | ↑ | ↓ |
| Smoking initiation | ↓ | ↓ | ↑ | | ↓ | ↓ | ↑ |
| Cigarettes smoked per day | ↑ | ↑ | ↑ | | ↑ | ↑ | ↑ |
| Alcoholic drinks per week | ↓ | ↓ | ↓ | | ↓ | ↓ | ↓ |
| Other exercises | | | | | ↓ | ↑ | ↓ |
| Heavy DIY | | | | | ↓ | ↑ | ↓ |
| Light DIY | | | | | ↑ | ↓ | ↑ |

| | |
|---|---|
| ↓ lower odds of urolithiasis (P > 0.05) | ↑ higher odds of urolithiasis (P > 0.05) |
| ↓ lower odds of urolithiasis (P < 0.05) | ↑ higher odds of urolithiasis (P < 0.05) |

**Fig 1. Overview of the associations of circulating biomarkers and lifestyle factors with urolithiasis.** IVW, inverse-variance weighted method; MRE, MR Egger method; WM, Weighted median method; SHBG, sex hormone-binding globulin; PTH, parathyroid hormone; CRP, c-reactive protein; IL6, interleukin-6; IL18, interleukin-18; IL27, interleukin-27; IL8, interleukin-8; IL16, pro-interleukin-16; IL1Ra, interleukin-1 receptor antagonist.

[10, 11]. The MR design was conducted according to three basic assumptions (**Fig 1**): the genetic variants selected as IVs should be robustly associated with the exposure (assumption 1); the selected variants should not be associated with any confounders (assumption 2); and the used variants should not be associated with the outcome, except by way of exposure

(assumption 3). The univariable MR analyses (UVMR) was used to evaluate the causal effect of lifestyle factors and circulating biomarkers upon urolithiasis, and multivariable MR analyses (MVMR) was applied to determine which exposure was causally associated with urolithiasis, independent of the other potentially relevant exposures. Compared with UVMR, MVMR allows for measured pleiotropy via any of the observed risk factors [13]. As our research involved re-analysis of publicly available summary-level data from large genome-wide association studies (GWAS), there was no requirement for additional ethical approval.

### Collection of urolithiasis-related GWAS data

Summary-level data on kidney stones/ureter stones/bladder stones/urolithiasis were sourced from UKB [14] and the FinnGen consortium. In the UKB cohort, 3,625 cases (459,308 controls) with kidney stones/ureter stones/bladder stones were defined by N20 in the International Classification of Diseases, 10th Revision (ICD-10), the Office of Population and Censuses Surveys, and self-reported operation codes. In the FinnGen consortium, 5,347 cases (213,445 controls) were defined by N20 in the ICD-10. The latest research has indicated that a sample overlap might bias the causal estimation in MR analysis [15, 16]. To minimize bias induced by a sample overlap, MR of urolithiasis (UKB) on exposure was not performed when exposure was derived from the UKB cohort. MR of urolithiasis (FinnGen) on exposure was not performed when exposure was derived from the FinnGen cohort. When exposure was derived from other cohorts, meta-analysis was applied to assess the combined effects (**Fig 1**).

### Collection of potential risk factors of urolithiasis and the corresponding GWAS data

We initially gathered potential risk factors linked to urinary stones from recent observational studies. After excluding factors already explored in the context of urolithiasis through Mendelian randomization, we ultimately identified 17 potential exposure factors for our subsequent analysis. These risk factors can be categorized into two groups, namely circulating biomarkers (estradiol levels, total testosterone levels, SHBG, parathyroid hormone (PTH), C-Reactive protein (CRP), Interleukin-6 (IL6), IL18, IL27, IL8, IL16, and Interleukin-1-receptor antagonist levels (IL1Ra)) and lifestyle factors (tea intake, garlic intake, coffee consumption, smoking, alcohol consumption, and physical activity). Subsequently, we obtained these publicly available summary-level data related to the 17 potential risk factors from large GWAS.

The summary-level statistical data on estradiol levels, total testosterone levels, and SHBG (N ≤ 425,097) were obtained from UK Biobank GWAS [17]. The summary-level statistical data on PTH (N = 3,301) were obtained from a large GWAS [18]. The summary-level statistical data on CRP were obtained from a meta-analysis of two GWAS, including 204,402 individuals of European ancestry [19]. The summary-level statistical data on circulating inflammatory biomarkers (N = 21,758) were obtained from a large GWAS [20]. The summary-level statistical data on tea intake and garlic intake (N ≤ 447,485) were obtained from UK Biobank GWAS [14]. The summary-level statistical data on coffee consumption was obtained from a large GWAS, including 375,833 European individuals [21]. The summary-level statistical data on the age of smoking initiation, smoking initiation, cigarettes smoked per day, and alcoholic drinks per week (N ≤ 607,291) were obtained from the GWAS and Sequencing Consortium of Alcohol and Nicotine use (GSCAN) [22]. The summary-level statistical data on types of physical activity in the last four weeks—heavy DIY (e.g., weeding, lawn mowing, carpentry, digging), light DIY (e.g., pruning, watering the lawn), and other exercises (e.g., swimming, cycling, keeping fit, bowling)—were obtained from UK Biobank GWAS [23].

## Genetic instrument of potential risk factors

A series of quality control steps were conducted to select eligible instrumental single-nucleotide polymorphisms (SNPs). First, SNPs robustly associated with exposure at the genome-wide significance level (ranging from P <5e−8) were selected. Second, we applied a linkage disequilibrium (LD) clumping process (LD r2 < 0.001, clumping distance cutoff = 10,000kb) to estimate the LD between SNPs. To minimize pleiotropy, Phenoscanner2 was applied to determine whether any exposure-associated SNPs were associated with relevant confounders of urolithiasis. Afterward, the SNPs that were unavailable in the outcome GWAS were removed, and extracting SNPs were not significantly associated with the outcome (P >5e−5). Moreover, the number of above-selected SNPs extracted from the outcome is greater than three. Furthermore, harmonization was applied in order to exclude palindromic and incompatible SNPs. Finally, to test whether there was a weak IV bias, namely genetic variants selected as IVs having a weak association with exposure, we calculated the F statistic ($F = R^2/ (1 − R^2) *(n − k − 1)/ k$; $R^2 = 2*MAF*(1-MAF) *Beta^2$; n, sample size; k, number of instrumental variables; and MAF, minor allele frequency) [24].

## Statistical analyses

The "TwoSampleMR" package based on R 4.10 was applied to perform MR analysis. After harmonization of the effect alleles across the GWAS of exposures and urolithiasis, the conventional fixed-effects inverse-variance weighted (IVW) method was conducted as the main statistical model of causal estimates. Estimates from the UKB and FinnGen cohorts were combined by meta-analysis method. A fixed-effects model was used where heterogeneity was low (when I2 values were lower than 50%); otherwise, a random-effects model was used. The IVW method assumed that instruments could affect the outcome only through the exposure of interest and not through any alternative pathway [25]. Moreover, MR-Egger and weighted-median methods were used to supplement IVW [26, 27]. The MR-Egger test for directional pleiotropy and Cochran's Q statistics were used to identify whether significant heterogeneity or directional pleiotropy was present. MVMR analyses were conducted using the "mv_multiple" function in the "TwoSampleMR" package.

## Results

### Information on outcomes and exposures

A total of two cohorts were included in our analysis. There were 5,347 (2.4%) individuals with urolithiasis in the FinnGen cohort and 3,625 (0.8%) patients with kidney stones/ureter stones/bladder stones in the UKB cohort (Table 1). The IVs of circulating biomarkers range from 4–395, and the genetic instruments of lifestyle factors range from 7–93 (Table 2). Nearly all of these exposures had strong genetic instruments (F statistics>10 for 18 of the 21 selected risk factors, Table 2). Detailed information on urolithiasis-independent SNPs (after the clumping process) for circulating biomarkers and lifestyle factors is listed in S1 and S2 Tables, respectively.

**Table 1. Description of urolithiasis outcomes used in this study.**

| Trait_name | Definition of CKD | N_case | N_control | GWAS_ID | Consortium |
|---|---|---|---|---|---|
| Kidney/ureter/bladder stone | N20 in ICD-10 | 3625 | 459308 | ukb-b-8297 | UKBB |
| Urolithiasis | N20 in ICD-10 | 5347 | 213445 | finn-b-N14_UROLITHIASIS | Finngen |

**Table 2. Characteristics of 17 reported risk factors were selected from PubMed.**

| Category | Phenotype_name | Abbreviation | GWAS ID | Sample Size | N_SNPs | F |
|---|---|---|---|---|---|---|
| **Circulating biomarkers** | Estradiol levels | - | ebi-a-GCST90012105 | 206927 | 13 | 10.66 |
| | Sex hormone-binding globulin levels | SHBG | ebi-a-GCST90012111 | 370125 | 395 | 19.35 |
| | Total testosterone levels | - | ebi-a-GCST90012114 | 425097 | 181 | 30.64 |
| | Parathyroid hormone | PTH | prot-a-2431 | 3,301 | 18 | 25.29 |
| | C-Reactive protein | CRP | ieu-b-4764 | 204402 | 57 | 258.46 |
| | Interleukin-6 levels | IL6 | ebi-a-GCST90012005 | 21758 | 14 | 73.06 |
| | Interleukin-18 levels | IL18 | ebi-a-GCST90012024 | 21758 | 8 | 176.50 |
| | Interleukin-27 levels | IL27 | ebi-a-GCST90012017 | 21758 | 13 | 281.15 |
| | Interleukin-8 levels | IL8 | ebi-a-GCST90011994 | 21758 | 16 | 56.88 |
| | Pro-interleukin-16 levels | IL16 | ebi-a-GCST90012049 | 21758 | 8 | 493.57 |
| | Interleukin-1-receptor antagonist levels | IL1Ra | ebi-a-GCST90012004 | 21758 | 4 | 262.52 |
| **Lifestyle Factors** | Tea intake | - | ukb-b-6066 | 447485 | 41 | 61.35 |
| | Garlic intake | - | ukb-b-17223 | 64949 | 19 | 10.06 |
| | Coffee intake | - | ukb-b-5237 | 428,860 | 12 | 273865.99 |
| | Age Of Smoking Initiation | - | ieu-b-24 | 341427 | 7 | 38.61 |
| | Smoking initiation | - | ieu-b-4877 | 607291 | 93 | 169.12 |
| | Cigarettes smoked per day | - | ieu-b-142 | 249752 | 23 | 349.70 |
| | Alcoholic drinks per week | - | ieu-b-73 | 335394 | 35 | 85.07 |
| | Types of physical activity in last 4 weeks: Other exercises (eg: swimming, cycling, keep fit, bowling) | - | ukb-b-8764 | 460376 | 15 | 9.56 |
| | Types of physical activity in last 4 weeks: Heavy DIY (eg: weeding, lawn mowing, carpentry, digging) | - | ukb-b-13184 | 460376 | 18 | 8.19 |
| | Types of physical activity in last 4 weeks: Light DIY (eg: pruning, watering the lawn) | - | ukb-b-11495 | 460376 | 13 | 9.99 |

## The UVMR analyses revealed the causal roles of five factors on urolithiasis

In the FinnGen consortium, higher genetically predicted estradiol levels, SHBG levels, tea intake, and other exercises (e.g., swimming, cycling, keeping fit, and bowling) were significantly associated with a lower risk of urolithiasis (**Fig 1**). The ORs of urolithiasis were 0.179 (95% CI, 0.042–0.751, P = 0.01877353) per one-SD increase in estradiol levels, 0.708 (95% CI, 0.555–0.903, P = 0.00544037) per one-SD increase in SHBG levels, 0.550 (95% CI, 0.345–0.878, P = 0.0121812) per one-unit increase in the log OR of tea intake, and 0.054 (95% CI, 0.008–0.363, P = 0.00269922) per one-unit increase in the log OR of other exercise (e.g., swimming, cycling, keeping fit, bowling) (**Fig 2A**). Intriguingly, we noticed that a higher genetically predicted IL6 level was significantly associated with an increased risk of urolithiasis (P = 0.0408157), supported by the MR-egger method (**Fig 1**). Although the statistical significance was inconsistent in the IVW method, the direction of the causal estimates remained the same with broadly comparable effect sizes (**Fig 1** and **S3 Table**). There was no causal association with other potential risk factors in the FinnGen cohort. After that, horizontal pleiotropy

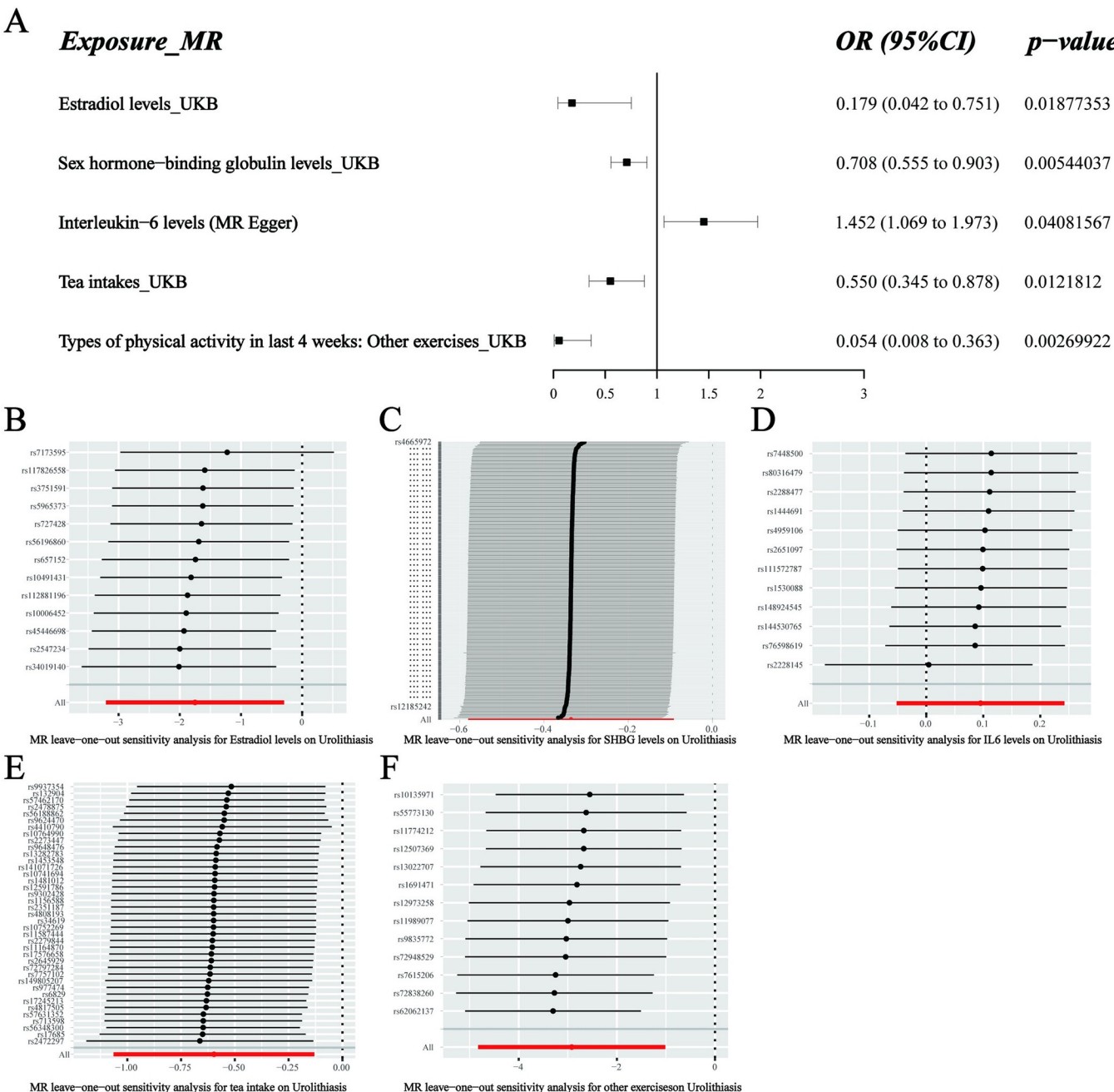

**Fig 2. Causal relationships of circulating biomarkers and lifestyle factors with the risk of urolithiasis in the FinnGen cohort.** A. Forest plot of Mendelian randomization results. Estimates were obtained using the inverse-variance weighted method. The boxes refer to the point estimates OR indicate the odds ratio, and the horizontal bars represent the 95% confidence interval. B–F. The MR leave-one-out sensitivity analysis for estradiol levels, SHBG levels, IL6 levels, tea intake, and types of physical activity in the last four weeks: other exercises (e.g., swimming, cycling, keeping fit, bowling) concerning urolithiasis.

results were evaluated using the intercept in MR-Egger regression, and the results detected no pleiotropy in all analyses (**S3 Table**). The leave-one-out sensitivity analysis did not identify any IV as an outlier (**Fig 2B–2F**).

In the UKB consortium, higher genetically predicted coffee consumption and alcoholic drinks per week were significantly associated with a lower risk of urolithiasis (**Fig 1**). The ORs

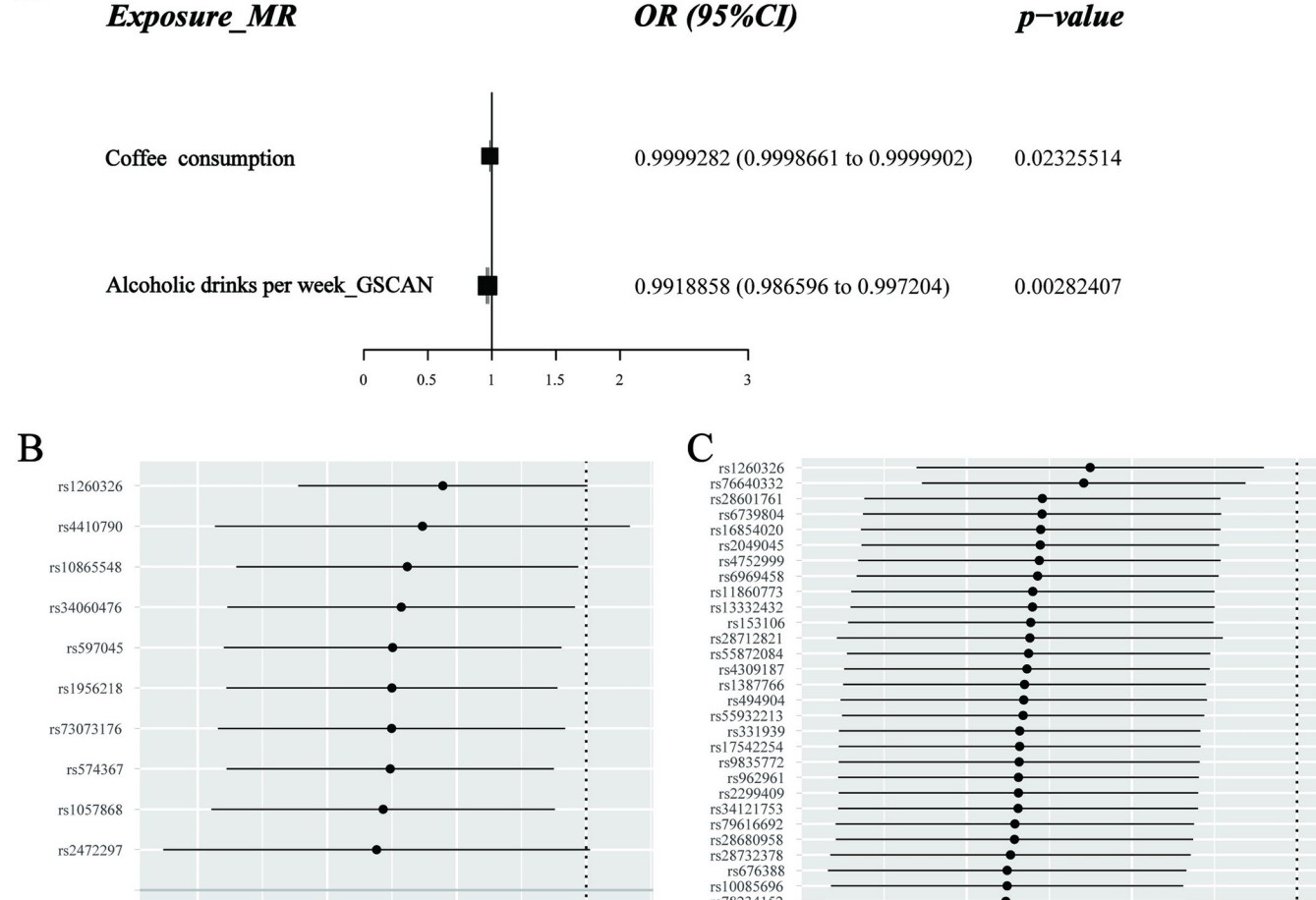

**Fig 3. Causal relationships of circulating biomarkers and lifestyle factors with the risk of urolithiasis in the UKB cohort.** A. Forest plot of Mendelian randomization results. B–C. The MR leave-one-out sensitivity analysis on urolithiasis concerning coffee consumption and alcoholic drinks per week.

of urolithiasis were 0.9999282 (95% CI, 0.9998661–0.9999902, P = 0.02325514) per one-unit increase in the log OR of coffee consumption and 0.9918858 (95% CI, 0.986596–0.997204, P = 0.00282407) per one-unit increase in the log OR of alcoholic drinks per week (**Fig 3A**). Moreover, higher genetically predicted levels of PTH, CRP, IL6, IL18, IL27, IL8, IL16, IL1Ra, age of smoking initiation, smoking initiation, and cigarettes smoked per day were not associated with urolithiasis in the UKB cohort. The robustness of the results in all sensitivity analyses suggests little to no interference by pleiotropy (P > 0.05, **S4 Table**). Furthermore, the leave-one-out analysis detected no outlier in this part (**Fig 3B and 3C**).

The meta-analysis results indicated that urolithiasis could be affected by alcoholic drinks per week (OR, 0.99187118, 95% CI, 0.98658174–0.99718898, P = 0.003), which was not discovered in the FinnGen cohort (**Fig 4A** and **S5 Table**). However, there was no causal association between other common risk factors (PTH, CRP, IL6, IL18, IL27, IL8, IL16, IL1Ra, coffee consumption, age of smoking initiation, smoking initiation, and cigarettes smoked per day) and

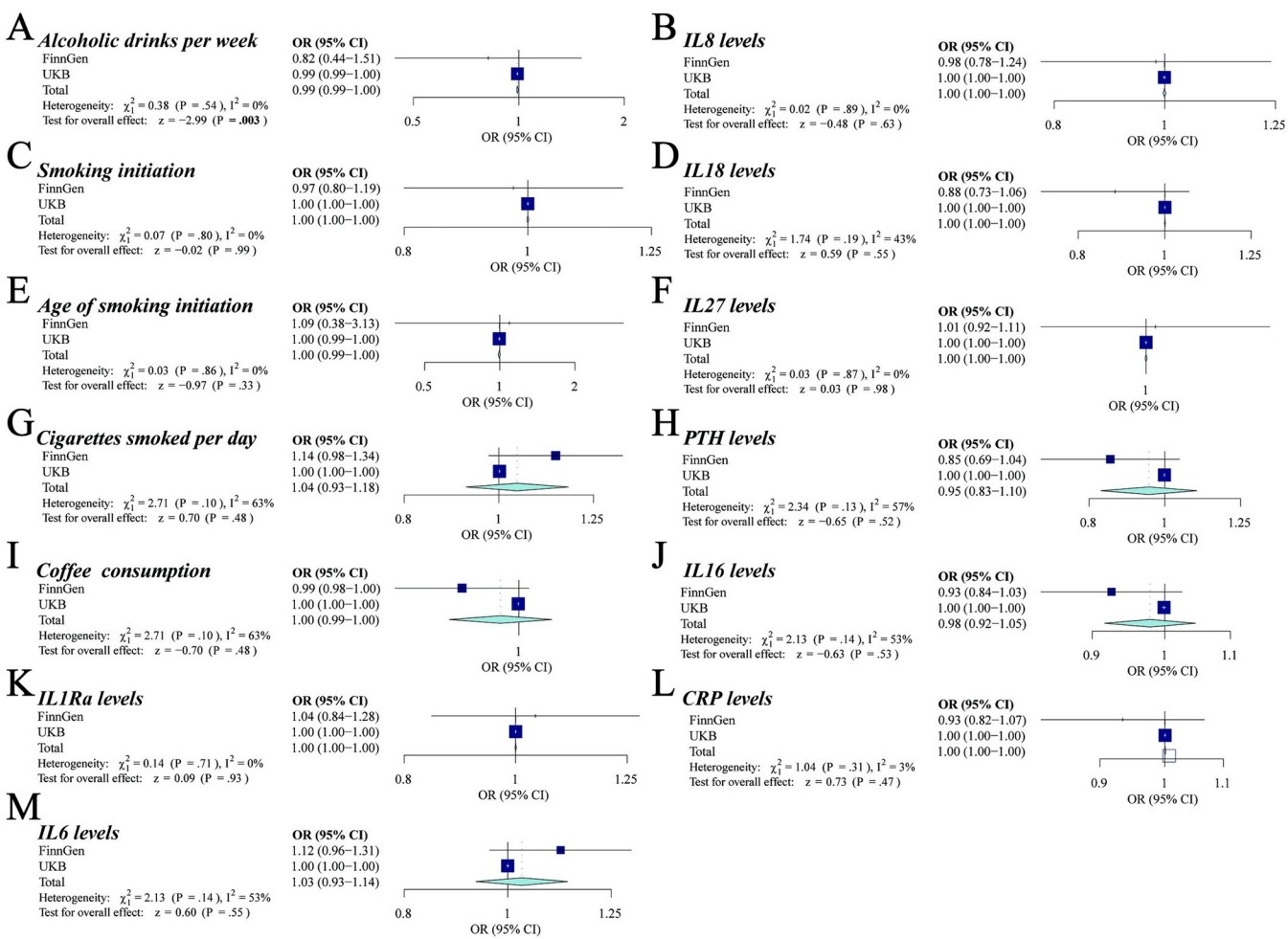

**Fig 4. Combined associations of genetically predicted circulating biomarkers and lifestyle factors with the risk of urolithiasis.**

urolithiasis in the meta-analysis (**Fig 4B–4M**). In conclusion, our MR findings support the causal roles of 5 factors upon urolithiasis.

## The MVMR analyses revealed the intricate interplay between these identified factors

To better understand the intricate interplay between these identified factors, we conducted Multivariate Mendelian Randomization (MVMR) analyses to comprehensive assessment their associations with urolithiasis (**Fig 5 and S6 Table**). In MVMR analyses, the causal association of estradiol on urolithiasis were still statistically significant after adjusting for tea intake (OR: 0.24; [95% CI: 0.09–0.67]) and other exercises (OR: 0.20; [95% CI: 0.05–0.75]). Similarly, the causal association of SHBG on urolithiasis remained significant after adjusting for tea intake (OR: 0.75; [95% CI: 0.58–0.97]), alcoholic drinks per week (OR: 0.75; [95% CI: 0.58–0.96]), and other exercises (OR: 0.70; [95% CI: 0.54–0.90]). Tea intakes also maintained its significant causal association with urolithiasis after adjusting for estradiol (OR: 0.51; [95% CI: 0.30–0.87]), SHBG (OR: 0.49; [95% CI: 0.25–0.94]), and other exercises (OR: 0.52; [95% CI: 0.31–0.89]). Additionally, the causal association between alcoholic

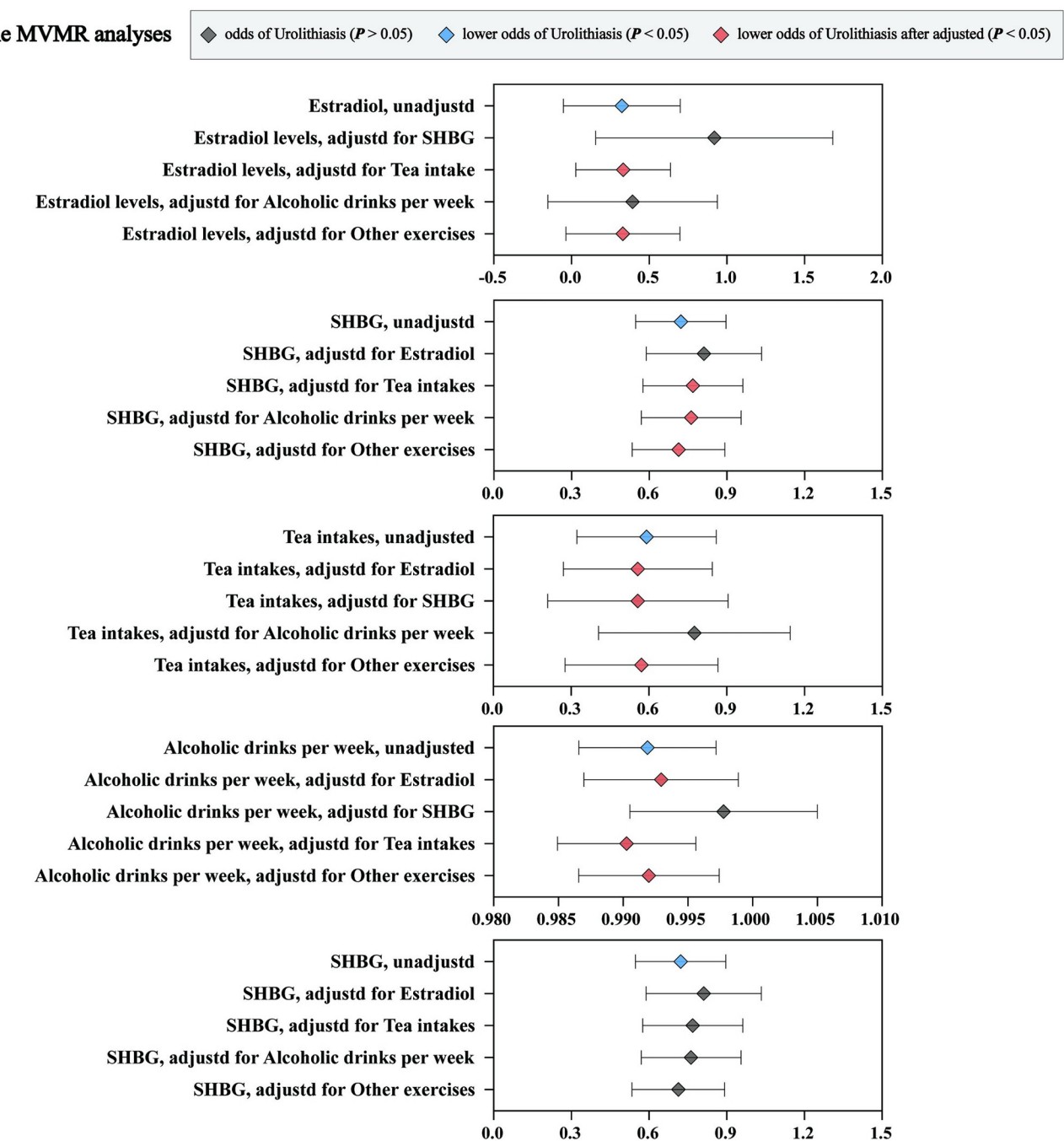

**Fig 5. Associations of genetically predicted risk factors with urolithiasis using multivariable MR analyses.**

drinks per week and urolithiasis remained significant after adjusting for estradiol (OR: 0.993; [95% CI: 0.987–0.999]), tea intake (OR: 0.990; [95% CI: 0.985–0.996]), and other exercises (OR: 0.992; [95% CI: 0.987–0.997]). However, after adjusting for these potential confounding variables, the previously observed causal association between other exercises and urolithiasis was no longer significant.

## Discussion

Revealing the independent causal link between risk factors and urolithiasis in large population datasets can yield valuable insights for disease prevention. In our study, the FinnGen and UKB cohorts were applied to identify the causal risk factors for kidney stones. Out of 21 potential factors, we established the causal roles of five, including two circulating biomarkers (SHBG and estradiol), and three lifestyle factors (tea consumption, weekly alcoholic drinks, and certain physical activities) through Univariate Mendelian Randomization (UVMR) analyses. However, after adjusting for potentially related exposures, the causal links between SHBG, estradiol, weekly alcohol consumption and urolithiasis did not prove independent of other factors. Only the associations of tea consumption and certain physical activities with urolithiasis remained. These findings may offer deeper understanding into the risk factors of renal stones and present potential strategies for their prevention.

In previous studies, the role of sex hormones has presented a controversial picture. An analysis of the National Health and Nutrition Examination Survey suggested no relationship between estradiol and renal stones in both males and female populations [28]. However, the incidence of female kidney stones is lower than age-matched males which was reported by Epidemiological data [29]. In addition, it was indicated that the higher remaining estradiol level in naturally postmenopausal women was a protective element against renal calcium oxalate stone, which is the most common kidney stone type [30]. The one major result from our UVMR analysis was that the high level of estradiol in human blood was causal linked to a low rate of kidney stone disease. Thus, our result strengthens the arguments for the protective role of estradiol in kidney stones formation. Also, estradiol could alter the cellular proteome of renal tubular cells, which decreased surface expression of CaOx crystal receptors, reduced intracellular metabolism, and enhanced cell proliferation and tissue healing, then prevent renal stone formation [31]. The association between SHBG and urolithiasis is another complex causal risk factor to consider. Contrary to our findings, Reza Naghii et al. reported a significantly increased in serum SHBG among kidney stone formers [6]. This discrepancy could potentially be attributed to various confounding factors. Notably, our MVMR analyses indicated that estradiol and SHGB are not independent causal factors of urolithiasis. In fact, high urinary oxalate excretion could be induced by testosterone and lead to calcium oxalate stones was indicated by animal studies [32, 33]. Most of the testosterone is bound to SHBG in the blood [34], thus we speculated that the increasing SHBG might indirectly explain the low level of testosterone in the blood. This evidence points to the possibility of intricate interactions among sex hormones, which collectively influence the formation of urolithiasis. Therefore, further work in the future is needed to detect the complicated mechanism behind it and the main targets of interventions.

Kidney stones are known to be associated with metabolic syndrome, including conditions such as obesity, diabetes, and hypertension [35]. Most metabolic syndrome is closely linked to the lifestyle of humans [36]. Thus, the lifestyle factors of renal stone patients, like diet and activity, also capture the attention of researchers [7, 37]. Fluid intake, a critical factor for kidney stone formation, could influence urinary chemical and physical properties [7]. As a popular beverages worldwide, tea consumption is an important culture within many countries. Because some tea is a source of stone components, tea intake is unrecommended amongst kidney stone formers [38, 39]. However, Shu et al. and Wang et al. reported that tea consumption was associated with a lower risk of urolithiasis [40, 41], especially green tea, which was indicated to have a protective function in kidney stone formation [8]. While black tea intake did not correlate with a decreased incidence of kidney stones, it did not elevate the risk either [42]. Our study lends further support to these findings by identifying tea consumption as a causal

factor associated with a lower risk of kidney stone formation. Several studies reported that hypercalciuria, the increasing production of uric acid, and renal oxidative stress induced by alcoholic drinking could lead to lithogenesis [43, 44]. However, some kinds of alcohol, like red wine, revealed an antioxidative effect on the host [45]. Furthermore, alcoholic drinking could dilute metabolites in urine and blood and promote urination [7]. These effects could reduce the risk of kidney stones [46]. Notably, our MR findings indicated that alcoholic drinks per week is associated with a reduced risk of kidney stone formation, but not directly responsible for it. These findings could contribute to dietary recommendations for kidney stone patients. Additionally, it was demonstrated that gut microbiota alteration could influence kidney stone formation [47]. What is more, diet, one of the most important factors with which to change gut microbiota [48], has to be noticed to investigate the specific mechanism of tea intake and alcoholic drinks in lithogenesis. On the other hand, a previous study depicted that a higher risk was found in marathon runners than in a normal population [49], which means that not all exercise benefits kidney stones. One explanation is that over-exercising could induce urine concentration and potentially promote crystal formation in urine [50]. Our results found a causal link between urolithiasis and some physical activity, including swimming, cycling, keeping fit, and bowling. The kidney stone patients who had undertaken such physical activity in the last 4 weeks were significantly associated with a lower incidence of urolithiasis. However, the previously observed causal association between other exercises and urolithiasis was no longer significant after adjusting for these factors. Therefore, the impact of physical activity on urolithiasis may be influenced by other confounding factors.

The present study integrated two nationwide biobanks and identified the independent causal roles of tea intake and some physical activity in urolithiasis. Although we conclude no causal associations concerning the remaining 15 markers (SHBG, estradiol, total testosterone levels, PTH, CRP, IL6, IL18, IL27, IL8, IL16, IL1Ra, garlic intake, coffee consumption, alcoholic drinks per week, and smoking), these negative results are as important as the positive discoveries for a fuller picture of the complicated etiology of urolithiasis. Inevitably, our research had certain limitations. First, similar to all MR studies, the potential influence of heterogeneity upon the results cannot be excluded entirely. Nevertheless, we are comforted by the fact that there was no evidence of pleiotropy in all MR analyses. Second, the diagnoses of diseases varied to some extent across the populations in our analyses. The FinnGen phenotypes were mainly based on digital health record data, while the UKB phenotypes predominantly relied on self-reported information, which might increase the potential for misreporting and underreporting. Third, further investigations should be carried out, taking into account the variations in sex hormones concentrations due to gender, menopausal status, and menstrual cycle phase. Lastly, our MR analyses have focused on the European population, thereby limiting further application to another population.

## Supporting information

**S1 Table. Genetic instruments of the 11 circulating risk factors in European ancestry.**
(XLSX)

**S2 Table. Genetic instruments of the 15 lifestyle risk factors.**
(XLSX)

**S3 Table. MR results of exposures on urolithiasis from the FinnGen.**
(XLSX)

**S4 Table. MR results of exposures on Kidney stone/ureter stone/bladder stone from the UK Biobank.**
(XLSX)

**S5 Table. Meta estimates of the causal associations of selected exposures on kidney stone.**
(XLSX)

**S6 Table. Meta estimates of the causal associations of multivariate mendelian randomization.**
(XLSX)

## Acknowledgments

We thank Home for Researchers editorial team (www.home-for-researchers.com) for language editing service.

## Author Contributions

**Conceptualization:** Hailin Fang.

**Formal analysis:** Jiwang Deng, Qingjiang Chen, Dong Chen, Bin Lai, Yongmao Zeng.

**Investigation:** Hailin Fang.

**Project administration:** Yuefu Han.

**Supervision:** Yuefu Han.

**Visualization:** Hailin Fang, Dong Chen, Pengfei Diao, Lian Peng.

**Writing – original draft:** Hailin Fang, Qingjiang Chen.

**Writing – review & editing:** Jiwang Deng.

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
