## [Decision Letter · Decision Letter 0]

26 Jun 2023

PONE-D-23-12318Univariable and multivariable mendelian randomization study revealed the modifiable risk factors of urolithiasisPLOS ONE

Dear Dr. Han,

Thank you for submitting your manuscript to PLOS ONE. After careful consideration, we feel that it has merit but does not fully meet PLOS ONE’s publication criteria as it currently stands. Therefore, we invite you to submit a revised version of the manuscript that addresses the points raised during the review process.

We look forward to receiving your revised manuscript.

Kind regards,

Sandar Tin Tin

Academic Editor

PLOS ONE

4. We note that Figure 1 in your submission contain copyrighted images. All PLOS content is published under the Creative Commons Attribution License (CC BY 4.0), which means that the manuscript, images, and Supporting Information files will be freely available online, and any third party is permitted to access, download, copy, distribute, and use these materials in any way, even commercially, with proper attribution. For more information, see our copyright guidelines: http://journals.plos.org/plosone/s/licenses-and-copyright.

b.If you are unable to obtain permission from the original copyright holder to publish these figures under the CC BY 4.0 license or if the copyright holder’s requirements are incompatible with the CC BY 4.0 license, please either i) remove the figure or ii) supply a replacement figure that complies with the CC BY 4.0 license. Please check copyright information on all replacement figures and update the figure caption with source information. If applicable, please specify in the figure caption text when a figure is similar but not identical to the original image and is therefore for illustrative purposes only.

Additional Editor Comments:

Please explain how the 17 risk factors were chosen, and why other important risk factors like obesity were not considered.Please consider the complex interplay between lifestyle factors and biomarkers more carefully in MVMR. Some biomarkers may mediate the association between lifestyle factors and urolithiasis, e.g., physical activity may reduce the risk of urolithiasis through its effect on body weight and/or sex steroid hormones. Please explain how MVMR was undertaken in the statistical analyses section, particularly including how the adjustment factors were chosen (e.g., for physical activity MVMR, explain why only tea intake and alcohol intake were adjusted but not other factors). Figure 1 should also reflect the complex interplay between lifestyle factors and biomarkers in influencing the risk of urolithiasis.Some of the biomarkers included are not proteins. For example, oestradiol and testosterone are steroids. Please clarify.Care should be taken in using oestradiol SNPs from UK Biobank as substantial variations in oestradiol concentrations by gender, by menopausal status in women and by menstrual cycle phase in pre-menopausal women were not accounted for in the GWAS. Additionally, the assay used to measure oestradiol in UK Biobank is not sensitive to detect very low concentrations in post-menopausal women; the majority of these women therefore had missing values for oestradiol. Please comment on the validity of the results related to oestradiol. Please define all acronyms (including those of biomarkers) when first used in the text.Some sentences are hard to read/interpret. Examples: "Due to such a re-analysis of publicly available summary-level data from large genome-wide association studies (GWAS)..."; "In MVMR analyses, estradiol had no independent causal effect on urolithiasis after adjusting for SHBG (IVW OR: 0.71; [95% CI: 0.28–1.77]). Similarly, the causal association of estradiol on urolithiasis did not remain with adjustment for SHBG (OR: 0.79; [95% CI: 0.60–1.04])". Please ensure that the language used is clear and unambiguous throughout the manuscript.

Reviewers' comments:

Reviewer's Responses to Questions

**Comments to the Author**

1. Is the manuscript technically sound, and do the data support the conclusions?

Reviewer #1: No

2. Has the statistical analysis been performed appropriately and rigorously? 

Reviewer #1: Yes

3. Have the authors made all data underlying the findings in their manuscript fully available?

Reviewer #1: Yes

4. Is the manuscript presented in an intelligible fashion and written in standard English?

Reviewer #1: Yes

5. Review Comments to the Author

Reviewer #1: The authors investigated publically available databases from the UK biobank and FinnGenn Consortium in order to construct Mendelian randomization models for risk factors for urinary stone disease. Data supported a role for sex hormone binding globulin as well as alcohol intake and tea intake and physical activity

1. In general, when constructing and instrument variable for a Mendelian randomization model, it is always best to choose steps in genetic variants that have known biological roles in the pathway of interest. This is because there are no great ways to absolutely prove assumptions 2 and 3 of Mendelian randomization models. Thus, ultimately, it is hard to say whether or not any of the lifestyle factors studied here is directly mediating the change in urinary stone risk that they are detecting and there models were if it is confounded by some other factor. For example, there could easily be confounding by body weight and/or obesity, a well studied risk factor for stone disease, for any of these lifestyle factors that they are studying here.

2. Use of this particular method to study sex hormone binding globulin however is more defenseable and of greater interest.

3. The authors do not specifically state how they chose the biomarkers or lifestyle factors that they included in this particular study. In particular many risk factors have been identified and they have chosen to study 17.

4. Figure 1 and Figure 2 do not add a lot to this particular paper in could be omitted.

6. PLOS authors have the option to publish the peer review history of their article (what does this mean?). If published, this will include your full peer review and any attached files.

Reviewer #1: No

---

## [Author Response · Author response to Decision Letter 0]

11 Jul 2023

Response to Reviewers

Dear Editor and reviewer:

On behalf of my co-authors, thank you for allowing a resubmission of our manuscript, with an opportunity to address the reviewers’ comments on our manuscript entitled “Univariable and multivariable mendelian randomization study revealed the modifiable risk factors of urolithiasis” (ID: PONE-D-23-12318). I really appreciate all your comments which are very helpful for improving the quality of our paper! We have studied the comments carefully and made corrections which we hope to meet with approval. The responds to the reviewers’ comments are as follows. 

Replies to the academic editor and reviewer’s comments:

Academic editor #: 

1. “Please explain how the 17 risk factors were chosen, and why other important risk factors like obesity were not considered.”

Response: 

Thank you for your insightful comments and suggestions. Regarding your question on how the 17 risk factors were selected and why other significant factors such as obesity were not considered, please allow us to clarify: We initiated our process by collating potential risk factors associated with urinary stones from recent observational studies. Subsequent to the exclusion of factors that had already been investigated in the context of urolithiasis through Mendelian randomization, we identified 17 potential exposure factors for our subsequent analysis. It is important to note that we did not include certain significant risk factors like obesity in our study because they have been extensively researched in previous studies. For instance, an increased risk of kidney stones associated with obesity has already been demonstrated in a Mendelian randomization analysis conducted by Yuan et al(1). Therefore, in this study, we chose to focus on risk factors that have not yet been explored using this method.

We hope this clarification is helpful and addresses your question. We appreciate your attention to this detail and welcome any further comments or suggestions you may have.

2. “Please consider the complex interplay between lifestyle factors and biomarkers more carefully in MVMR. Some biomarkers may mediate the association between lifestyle factors and urolithiasis, e.g., physical activity may reduce the risk of urolithiasis through its effect on body weight and/or sex steroid hormones. Please explain how MVMR was undertaken in the statistical analyses section, particularly including how the adjustment factors were chosen (e.g., for physical activity MVMR, explain why only tea intake and alcohol intake were adjusted but not other factors). Figure 1 should also reflect the complex interplay between lifestyle factors and biomarkers in influencing the risk of urolithiasis.”

Response: 

Thanks for your comment regarding the complex interplay between lifestyle factors and biomarkers in the context of Multivariate Mendelian Randomization (MVMR). Based on your valuable feedback, we conducted MVMR analyses to further investigate the intricate relationships between the identified lifestyle factors and biomarkers, and their associations with urolithiasis. Our findings revealed that the causal association between estradiol and urolithiasis remained statistically significant even after adjusting for tea intake and other exercises. Similarly, the causal association between SHBG and urolithiasis remained significant after adjusting for tea intake, alcoholic drinks per week, and other exercises. Furthermore, tea intake maintained its significant causal association with urolithiasis even after adjusting for estradiol, SHBG, and other exercises. Additionally, the causal association between alcoholic drinks per week and urolithiasis remained significant after adjusting for estradiol, tea intake, and other exercises. However, after adjusting for these potential confounding variables, the previously observed causal association between other exercises and urolithiasis was no longer significant. Furthermore, we noted that the causal relationship between estradiol and urolithiasis may be influenced by the causal association between SHBG and urolithiasis, indicating a potential interplay between these two biomarkers.

In the statistical analyses section, we provide a detailed description of how Multivariate Mendelian Randomization (MVMR) was conducted. We appreciate your feedback, and we have incorporated these findings and considerations into our response to enhance the clarity and conciseness of the manuscript.

3. “Some of the biomarkers included are not proteins. For example, oestradiol and testosterone are steroids. Please clarify.”

Response: 

Thanks for your insightful comments and suggestions. You are correct that not all biomarkers included in the study are proteins. Oestradiol and testosterone are indeed steroids, which are important hormonal biomarkers. They play significant roles in various physiological processes, including the regulation of reproductive functions and the development of secondary sexual characteristics.

In the context of our study, we considered oestradiol and testosterone as biomarkers due to their relevance in urolithiasis and their potential associations with kidney stone formation. Although they are not proteins, they are important molecules that can provide insights into the hormonal factors that may contribute to the development of kidney stones.

Thank you for pointing out this distinction, and we have made this clarification in the manuscript to accurately represent the biomarkers used in our study.

4. “Care should be taken in using oestradiol SNPs from UK Biobank as substantial variations in oestradiol concentrations by gender, by menopausal status in women and by menstrual cycle phase in pre-menopausal women were not accounted for in the GWAS. Additionally, the assay used to measure oestradiol in UK Biobank is not sensitive to detect very low concentrations in post-menopausal women; the majority of these women therefore had missing values for oestradiol. Please comment on the validity of the results related to oestradiol.”

Response: 

We appreciate the Editor's concern regarding the use of estradiol SNPs from the UK Biobank. Indeed, variations in estradiol concentrations due to gender, menopausal status, and menstrual cycle phase in pre-menopausal women were not taken into account in the Genome-Wide Association Study (GWAS). Furthermore, the sensitivity of the assay used to measure estradiol in the UK Biobank may not be optimal for detecting very low concentrations in post-menopausal women, resulting in missing values for a majority of these individuals. We recognize these issues and agree that they pose significant challenges to the validity of the results related to estradiol. Therefore, we have interpreted our findings on estradiol with caution.

However, it's important to note that Mendelian Randomization (MR) inherently assumes that the genetic variants (in this case, estradiol SNPs) used as instrumental variables are robustly associated with the exposure (estradiol levels). The strength of the MR analysis does not necessarily depend on the granularity of the exposure measurement (i.e., varying estradiol levels across different physiological states) but rather on the robustness of the genetic association. Therefore, while the missing values for post-menopausal women and unaccounted variations are limitations, they do not invalidate the utility of estradiol SNPs in our MR analysis.

Nonetheless, we agree that further investigations should be carried out, taking into account the variations in estradiol concentrations due to gender, menopausal status, and menstrual cycle phase. We have included this point in the discussion section to emphasize the need for future research in this area. This would provide more accurate and comprehensive insights into the role of estradiol in urolithiasis.

5. “Please define all acronyms (including those of biomarkers) when first used in the text.”

Response: 

Thank you for your insightful suggestion. We understand the importance of clarity and accessibility in our manuscript. Therefore, we have ensured to define all acronyms, including those of biomarkers, when they are first introduced in the text to make our manuscript easier to understand for a broad readership. We appreciate your patience and understanding in this matter and apologize for any confusion caused previously.

6. “Some sentences are hard to read/interpret. Examples: "Due to such a re-analysis of publicly available summary-level data from large genome-wide association studies (GWAS)..."; "In MVMR analyses, estradiol had no independent causal effect on urolithiasis after adjusting for SHBG (IVW OR: 0.71; [95% CI: 0.28–1.77]). Similarly, the causal association of estradiol on urolithiasis did not remain with adjustment for SHBG (OR: 0.79; [95% CI: 0.60–1.04])". Please ensure that the language used is clear and unambiguous throughout the manuscript.”

Response: 

Thank you for your insightful comments and for highlighting those areas where the clarity and readability of our manuscript could be improved. We greatly appreciate the specific examples you provided, as they were instrumental in guiding our revisions.

We have carefully gone through the entire manuscript and made extensive edits to enhance the precision and clarity of our language, ensuring it communicates our findings as accurately and straightforwardly as possible. We have paid particular attention to the sentences you pointed out and rephrased them accordingly.

We trust these improvements have adequately addressed your concerns, and we are hopeful that the revised manuscript will now meet the standards of the journal. We believe these changes not only enhance the manuscript's readability but also its overall quality.

We once again express our gratitude for your valuable input, which has significantly contributed to the refinement of our manuscript.

Reviewer#: 

1. “In general, when constructing and instrument variable for a Mendelian randomization model, it is always best to choose steps in genetic variants that have known biological roles in the pathway of interest. This is because there are no great ways to absolutely prove assumptions 2 and 3 of Mendelian randomization models. Thus, ultimately, it is hard to say whether or not any of the lifestyle factors studied here is directly mediating the change in urinary stone risk that they are detecting and there models were if it is confounded by some other factor. For example, there could easily be confounding by body weight and/or obesity, a well studied risk factor for stone disease, for any of these lifestyle factors that they are studying here.” And “The authors do not specifically state how they chose the biomarkers or lifestyle factors that they included in this particular study. In particular many risk factors have been identified and they have chosen to study 17.”

Response: 

Thank you for your valuable comments and insightful questions. Regarding your concerns about the construction of an instrument variable for Mendelian randomization models and the potential confounding effects, we completely agree. It is indeed ideal to select genetic variants with known biological roles in the pathway of interest. For this study, we aimed to conduct a broad exploratory analysis by considering a wider range of potential risk factors, including lifestyle factors and biomarkers that have been associated with urinary stone risk in observational studies, but not yet thoroughly explored using Mendelian randomization. This selection was not intended to exclude other significant factors.

On the subject of confounding by body weight or obesity, we appreciate the importance of this point. In fact, the relationship between obesity and kidney stone disease has been well-documented, including in a previous Mendelian randomization analysis conducted by Yuan et al. As such, obesity was not included as a factor in our current study as we sought to investigate novel potential risk factors for urolithiasis. We understand that our chosen factors might be influenced by other factors such as obesity. We believe our study contributes to the ongoing exploration of the multifaceted etiology of urinary stones, and future studies could integrate the findings of our study with factors like obesity to provide a more comprehensive picture.

As for the selection of the 17 factors, we based this on recent observational studies and excluded factors that have already been extensively researched via Mendelian randomization in the context of urolithiasis. Our aim was to cast a wider net to potentially uncover less-explored risk factors for urinary stones.

We appreciate your feedback, which will help us refine and improve our research. We will certainly take these concerns into consideration in our future research efforts.

2. “Use of this particular method to study sex hormone binding globulin however is more defenseable and of greater interest.”

Response: 

Thank you for your positive remark on our analysis of sex hormone binding globulin (SHBG). We concur that this is a particularly interesting factor in the context of urinary stones.

Our investigation into SHBG was guided by previous studies suggesting a potential link between sex hormones and urinary stone risk. As SHBG binds to sex hormones, we hypothesized that it could play a significant role in urolithiasis.

We believe that by using Mendelian randomization, we have the potential to minimize the biases often seen in observational studies, and thus can provide more reliable and robust evidence of SHBG's role in urinary stone disease.

We appreciate your feedback and are encouraged by your interest in this part of our study.

3. “Figure 1 and Figure 2 do not add a lot to this particular paper in could be omitted.”

Response: 

Thank you for your feedback regarding Figures 1 and 2. We appreciate your critical review and perspective. In response, we have decided to remove Figure 1 in the revised manuscript to streamline the presentation of our results.

However, we believe that Figure 2 plays an essential role in demonstrating the causal effects and statistical significance of various exposure factors on urolithiasis in the FinnGen and UK Biobank consortia using three common Mendelian randomization methods. This figure provides a visual overview and facilitates comprehension of our key findings.

We understand that the complexity of the results presented in Figure 2 may not immediately appear essential, but we feel it significantly contributes to the overall interpretation and understanding of the data.

We hope this clarification helps, and we appreciate your understanding.

1. Yuan S, Larsson SC. Assessing Causal Associations of Obesity and Diabetes with Kidney Stones Using Mendelian Randomization Analysis. Mol Genet Metab (2021) 134(1-2):212-5. doi: 10.1016/j.ymgme.2021.08.010.

---

## [Decision Letter · Decision Letter 1]

8 Aug 2023

Univariable and multivariable mendelian randomization study revealed the modifiable risk factors of urolithiasis

PONE-D-23-12318R1

Dear Dr. Han,

We’re pleased to inform you that your manuscript has been judged scientifically suitable for publication and will be formally accepted for publication once it meets all outstanding technical requirements.

Kind regards,

Sandar Tin Tin

Academic Editor

PLOS ONE

Reviewers' comments:

Reviewer's Responses to Questions

**Comments to the Author**

1. If the authors have adequately addressed your comments raised in a previous round of review and you feel that this manuscript is now acceptable for publication, you may indicate that here to bypass the “Comments to the Author” section, enter your conflict of interest statement in the “Confidential to Editor” section, and submit your "Accept" recommendation.

Reviewer #1: All comments have been addressed

2. Is the manuscript technically sound, and do the data support the conclusions?

Reviewer #1: Yes

3. Has the statistical analysis been performed appropriately and rigorously? 

Reviewer #1: Yes

4. Have the authors made all data underlying the findings in their manuscript fully available?

Reviewer #1: Yes

5. Is the manuscript presented in an intelligible fashion and written in standard English?

Reviewer #1: Yes

6. Review Comments to the Author

Reviewer #1: Overall improved. No further comments at this time.

7. PLOS authors have the option to publish the peer review history of their article (what does this mean?). If published, this will include your full peer review and any attached files.

Reviewer #1: No

---

## [Editor Report · Acceptance letter]

18 Aug 2023

PONE-D-23-12318R1 

Univariable and multivariable mendelian randomization study revealed the modifiable risk factors of urolithiasis 

Dear Dr. Han:

I'm pleased to inform you that your manuscript has been deemed suitable for publication in PLOS ONE. Congratulations! Your manuscript is now with our production department. 

Kind regards, 

on behalf of

Dr. Sandar Tin Tin 

Academic Editor

PLOS ONE